# Is the Attention Matrix Really the Key to Self-Attention in Multivariate Long-Term Time Series Forecasting?

## Abstract

In multivariate long-term time series forecasting, the success of self-attention is commonly attributed to the attention matrix that encodes token interactions. In this paper, we provide evidence that challenges this view. Through extensive experiments on three classic and three latest Transformer models, we find that dot-product attention can be replaced by element-wise operations without token interaction, such as the addition and Hadamard product, while maintaining or even improving accuracy. This motivates our central hypothesis: the effectiveness of self-attention in this task arises not from the dynamic attention matrix, but from the multi-branch feature extraction enabled by the parallel Query, Key, and Value projections and their fusion. To validate this hypothesis, we construct a minimalist multi-branch MLP that isolates the 'multi-branch mapping with element-wise operation' structure from the Transformer and show that it achieves competitive performance. Our findings indicate that the source of performance in self-attention is often misinterpreted, as its actual advantage stems from the architectural principle of multi-branch mapping and fusion, rather than the attention matrix. Anonymous code is available at: https://anonymous.4open.science/r/Attention-01F4/

## 1 Introduction

Multivariate long-term time series forecasting (MLTSF) is a fundamental problem across critical domains such as energy management Hussein & Awad (2024), economic forecasting Li et al. (2024), and meteorology Chen et al. (2023). In recent years, Transformer-based models, driven by their core self-attention (SA) mechanism, have become the standard benchmark in this field. From PatchTST and iTransformer to the latest state-of-the-art (SOTA) models, they have achieved a dominant position in major benchmarks. These models are widely believed to owe their success to the self-attention mechanism, specifically the dynamic computation of token-to-token dependencies via the attention matrix.

However, is the attention matrix really the key as we believe? Our central claim in this paper is straightforward: the source of effectiveness for self-attention in multivariate long-term time series forecasting may be profoundly misunderstood. Through extensive ablation studies, we demonstrate that the primary contributor to the performance is not the attention matrix, but rather the inherent multi-branch feature extraction architecture that projects inputs into the parallel Query, Key, and Value (QKV) matrices and fusion. Our research reveals that replacing its dynamic token-interacting calculation with simple element-wise operations, such as the Hadamard product and addition, results in no significant loss in accuracy, and in many cases, even leads to improvements.

To substantiate our claim, we conduct systematic ablation studies on three classic and three latest Transformer models. They are based on vanilla self-attention, which can eliminate the interference of human factors in the ablation studies. We first reproduce the baseline performance of these models, then show that our simplified versions, which replace the dynamic token-interacting calculation with simple element-wise operations, can achieve comparable or even better results. Finally, we construct a purpose-built multi-branch MLP model that retains only the QKV mapping and simple element-wise operations, proving it is sufficient on its own to achieve competitive performance.

To be clear, our goal is not to entirely dismiss the immense success of Transformers or SA in other domains, such as natural language processing. In contrast, we aim to precisely analyze the role of this powerful mechanism in multivariate long-term time series forecasting. We hope to reveal the true drivers of its success, thereby helping the research community avoid an uncritical pursuit of increasingly complex attention variants and pointing the way toward genuinely efficient model design.

Our contributions can be summarized as follows.

- Through extensive experiments on six classic and latest models, we provide strong evidence that the attention matrix is not essential for MLTSF. We show it can be replaced by simple element-wise operations while maintaining or even improving performance.
- Based on this finding, we propose a core hypothesis that the source of self-attention's effectiveness has been misattributed. We argue the true driver is not the attention matrix, but the inherent multi-branch mapping and fusion architecture that projects inputs into Q, K, and V representations.
- We validate this hypothesis by constructing a purpose-built multi-branch MLP that isolates this principle. The competitive performance of this minimalist model proves that the architecture is sufficient on its own, providing decisive evidence for our hypothesis.

## 2 RELATED WORK

This section builds a focused narrative around the use and understanding of attention mechanisms in MLTSF, identifying the consensus, emerging alternatives, and the specific gap our work fills. Unlike standard literature reviews, our goal is not to list methods but to sharpen the reader's understanding of what has been believed, what has been questioned, and what has been left unexplored.

### 2.1 SELF-ATTENTION AS THE CORNERSTONE OF MLTSF

The recent dominance of Transformer-based architectures in MLTSF is rooted in the assumption that the attention matrixformed via matrix multiplication between Query and Key, is believed to be essential for modeling long-range dependencies. The attention matrix computes the correlation between tokens. Different Transformer variants adopt diverse tokenization strategies: (i) Time-token attention: Informer, Autoformer, FEDformer group all variable values at each time step into a token. (ii) Patch-token attention: PatchTST Nie et al. (2022) slices each univariate series into patches, treating each as a token. (iii) Variate-token attention: iTransformer Liu et al. (2023) treats each variable as a token to model cross-variable dependencies. (iii) Hybrid attention: TimePFN Taga et al. (2025), ICTSP Lu et al. (2025), Leddam Yu et al. (2024), and other methods Huang et al. (2024); Xue et al. (2023); Chen et al. (2024); Zhang & Yan (2023); Wang et al. (2024a; 2025) combine time and variate tokens. Despite their diversity, these models converge on the belief that performance gains come from the dynamic token interaction enabled by the attention matrix.

### 2.2 THE RISE OF SIMPLER ALTERNATIVES

The success of attention-free models like DLinear Zeng et al. (2023), PatchMLP Tang & Zhang (2025), and FITS Xu et al. (2024) has prompted a reassessment. These models, though devoid of any attention mechanism, perform competitively across benchmarks. CATS Kim et al. (2024) explored the effectiveness of the self-attention mechanism by removing it entirely. However, they do not attempt to dissect which parts of the attention module contribute most, nor whether certain internal architectural motifs, like the QKV projections with fusion, may carry predictive value independently.

### 2.3 REVISITING THE ARCHITECTURE: MULTI-BRANCH FEATURE EXTRACTION

Every self-attention block begins with projecting the input into three independent branches: Query, Key, and Value. While these branches are usually seen as precursors to attention computation, they also constitute a fixed multi-branch architecture that performs feature extraction. This structure is present in all Transformer variants but has received no attention as a potential source of effectiveness on its own.

This motivates our central question: in MLTSF, does the performance of self-attention stem from the dynamic attention matrix, or from the upstream QKV projection? Our study is the first to isolate and examine this architectural divide, offering a new lens for interpreting the success of Transformer-based forecasting models.

## 3 EXPERIMENTAL SETUP

To systematically evaluate the true role of the attention matrix in the self-attention mechanism for MLTSF, we design a series of rigorous experiments. This chapter details the baseline models we select, the structural ablation methods we propose, the specialized model used for hypothesis validation, and the datasets and evaluation criteria employed in our study. All experimental settings for the ablation studies are completely consistent with the corresponding baselines. The experimental settings of MB-MLP are consistent with the well-known baseline 'PatchTST' to eliminate human interference. The specific implementation details are in the Appendix A.3.

### 3.1 BASELINES

We select 6 representative and publicly available Transformer models in the MLTSF domain as baselines. These models cover different application paradigms of vanilla self-attention, allowing for a broad validation of our findings' universality. They are divided into two groups:

#### CLASSIC FOUNDATIONAL MODELS

- **Informer-like Model Zhou et al. (2021):** Represents the classic time-token paradigm. Our implementation is based on the Informer architecture, but utilizes the vanilla self-attention mechanism to serve as a foundational baseline.

- **PatchTST Nie et al. (2022):** Employs the patch-token paradigm, treating subsequences of each variable as tokens, and is one of the current top-performing baselines.

- **iTransformer Liu et al. (2023):** Utilizes the variate-token paradigm, treating the entire sequence of each variable as a token to capture inter-variable dependencies. We do not target the temporal dependency paradigm but instead explore various forms of self-attention paradigms.

#### LATEST MODELS

- **TimePFN (AAAI 25) Taga et al. (2025), ICTSP (ICLR 25) Lu et al. (2025), Leddam (ICML 24) Yu et al. (2024):** Represents the latest models published in 2025 and 2024 that employ vanilla self-attention with hybrid paradigm, often combining time- and variate-dimension tokenization. Since there are very few recently published SOTA models that apply vanilla self-attention (most are modified self-attention or cross-attention), we carefully select the above 3 most suitable ones as baselines.

In reproducing these models, we strictly adhere to the hyperparameters and training settings recommended in their original papers and official codebases to ensure a fair comparison. Since TimePFN Taga et al. (2025) only provides the ckpt for inference, we conduct experiments in the form of inference. In addition, as ICTSP Lu et al. (2025) only provides the scripts for the ETT dataset, we can only report the results of ETT. Due to the extremely high time cost of Leddam Yu et al. (2024), for large-scale Traffic and Electricity datasets, we can only conduct experiments and report at a prediction step length of 96.

### 3.2 PROPOSED STRUCTURAL ABLATIONS

As shown in Figure 1, to isolate and examine the respective contributions of the QKV multi-branch mapping and the subsequent attention matrix calculation, we propose two structural ablation modifications to the core SA module of all the aforementioned reference models. Our goal is to replace the core matrix multiplication for attention matrix with simpler, element-wise operations.

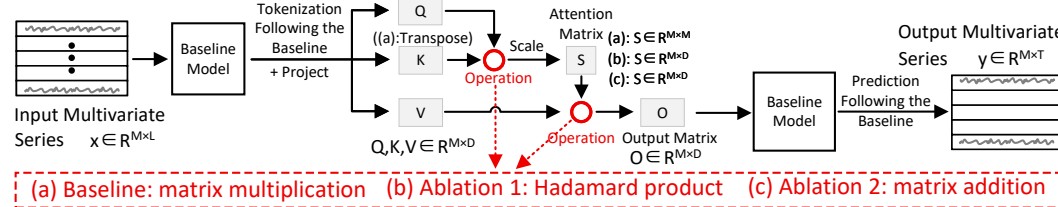

Figure 1: A schematic of our proposed structural ablation methods. The red circle represents the matrix operation to be ablated. Specifically, (a), (b), and (c) within the red box respectively represent matrix multiplication, Hadamard product, and addition. For ease of understanding, we take iTransformer baseline as an example to label the shape of the tensor. In practice, the settings of different baselines should be followed.

- **Standard Self-Attention (Baseline 1):** The original SA calculation, that is, $\text{Attention}(\mathbf{Q}, \mathbf{K}, \mathbf{V}) = \text{Softmax}(\frac{\mathbf{Q}\mathbf{K}^\top}{\sqrt{d_k}})\mathbf{V}$.

- **Sparse Self-Attention (Baseline 2):** We also test the sparse self-attention mechanism (ProbAttention) to verify the universality of our findings.

- **Attention-to-Hadamard (Ablation 1):** We replace the matrix multiplication of the two baselines with the Hadamard product (element-wise multiplication). For example, the calculation for baseline 1 becomes $\text{Attention}(\mathbf{Q}, \mathbf{K}, \mathbf{V}) = \text{Softmax}(\frac{\mathbf{Q}\odot\mathbf{K}}{\sqrt{d_k}}) \odot V$.

- **Attention-to-Addition (Ablation 2):** We replace the matrix multiplication of the two baselines with element-wise addition. For example, the calculation for baseline 1 becomes $\text{Attention}(\mathbf{Q}, \mathbf{K}, \mathbf{V}) = \text{Softmax}(\frac{\mathbf{Q}+\mathbf{K}}{\sqrt{d_k}}) + V$.

The subtlety of these modifications is that they fully preserve the "multi-branch" characteristic of the model, where the input is projected into three independent, trainable branches (Q, K, V), but completely remove the dynamic, token-interaction-dependent attention weight matrix, which is considered the soul of SA. It should be noted that each row of the ablated attention matrix represents the internal information of each token rather than interactive information, so Softmax will not introduce interactions between tokens. If the model's performance remains stable after these changes, it provides strong evidence for our core hypothesis.

### 3.3 VALIDATION MODEL: MB-MLP

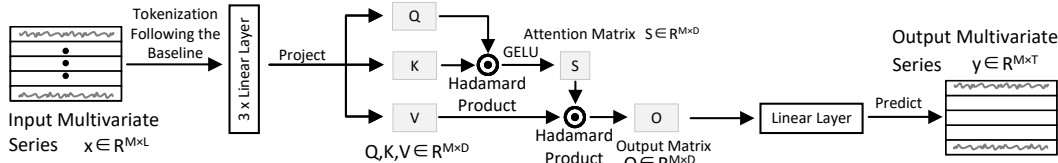

Figure 2: A schematic of our proposed minimalist validation model named Multi-Branch MLP (MB-MLP). It only retains the core that we want to validate, namely multi-branch feature extraction and fusion. For ease of understanding, we take iTransformer baseline as an example to label the shape of the tensor. In practice, the settings of different baselines should be followed.

To further validate our core hypothesis, that model performance primarily stems from the QKV multi-branch mapping architecture, we design and build a minimalist validation model named Multi-Branch MLP (MB-MLP), shown in Figure 2. It completely adopts the hyperparameters of the baseline 'PatchTST' without any adjustment or optimization to eliminate the influence of hyperparameters.

MB-MLP is intended to completely isolate and strip down to the core components we believe are truly effective, away from the complex Transformer architecture. Its workflow is as follows:

1. **Tokenization:** We follow the tokenization paradigm of the baselines; for example, when comparing with the Informer-like model, follow its time-token paradigm. Therefore, the shape of the tokenization result is the same as that of the baselines.

2. **Multi-Branch Mapping:** Like standard SA, the tokenization result is projected into Q, K, and V representations through three independent linear layers in parallel. The shape of Q, K, and V is the same as that of the baselines.

3. **Simple Fusion:** We use the same element-wise operations, Hadamard product, as in our ablation studies to fuse the features of Q, K, and V.

4. **Non-linear Activation Function:** We conduct an ablation on the Softmax in SA, verifying the effectiveness of non-linear activation (detailed results are in the Appendix). Therefore, MB-MLP needs to retain it and apply GELU to introduce nonlinearity to the product of Q and K. The shape of the obtained output matrix is the same as that of the baselines.

5. **Prediction Head:** We follow the settings of the baselines, that is, applying a linear layer as the prediction head to obtain the output multivariate series.

If this minimalist model can achieve near performance to vanilla SA models, such as PatchTST Nie et al. (2022) and iTransformer Liu et al. (2023), it would provide decisive evidence for the argument that the QKV multi-branch mapping and fusion is the key source of self-attention, rather than the attention matrix.

### 3.4 DATASETS AND EVALUATION METRICS

Table 1: Benchmark datasets employed in baselines' papers.

| Datasets | Traffic | Electricity | Solar | Weather | ETTh1 | ETTh2 | ETTm1 | ETTm2 |
|---|---|---|---|---|---|---|---|---|
| Variates | 862 | 321 | 137 | 21 | 7 | 7 | 7 | 7 |
| Timesteps | 17544 | 26304 | 52560 | 52696 | 17420 | 17420 | 69680 | 69680 |
| Granularity | 1hour | 1hour | 10min | 10min | 1hour | 1hour | 5min | 5min |

To maintain consistency with each baseline model, we use the datasets from their respective papers. The consolidated benchmark datasets are as follows: Traffic, Electricity, Solar, Weather, ETTh1, ETTh2, ETTm1, ETTm2. Their statistics are shown in Table 1 and details are in the Appendix A.2.

In MLTSF, we represent the input multiple time series as $x \in R^{M \times L}$, where $M$ is the number of variate and $L$ is the size of look-back window. For each single series of $i$-$th$ variate $x^{(i)} = (x_1^{(i)}, \ldots, x_L^{(i)}) \in R^{1 \times L}$, where $i = 1, \ldots, M$, the goal is to forecast $T$ future values $y^{(i)} = (x_{L+1}^{(i)}, \ldots, x_{L+T}^{(i)}) \in R^{1 \times T}$. We represent the multivariate prediction result as $y \in R^{M \times T}$. Following baseline models, we use the Mean Squared Error (MSE) and Mean Absolute Error (MAE) as metrics to evaluate the predictive performance of the models. The prediction horizons are $\{96, 192, 336, 720\}$.

## 4 RESULTS AND ANALYSIS

In this section, we provide the details of our comprehensive evaluation. Specifically, we ask the following research questions: (RQ1) Is the attention matrix really the key to self-attention in MLTSF? (RQ2) Does the performance of self-attention primarily stem from its multi-branch feature mapping architecture? (RQ3) Is the attention matrix worth the computational cost?

### 4.1 RQ1: IS THE ATTENTION MATRIX REALLY THE KEY TO SELF-ATTENTION IN MLTSF?

To answer this question, we conduct a series of structural ablation studies on 6 baseline models. We first present a global visual comparison (average error across four prediction horizon), with the criterion being whether the compared error curves have an overall deviation, as shown in Figure 3. Except for two significant deviation points, all other error curves almost overlap, indicating that removing the dynamic attention matrix has no significant impact. Moreover, in some intervals, the error curves of the ablation schemes (red and green) are even below the baseline, further validating the conclusion. Then, we explain the ablation study of each baseline one by one as follows. We

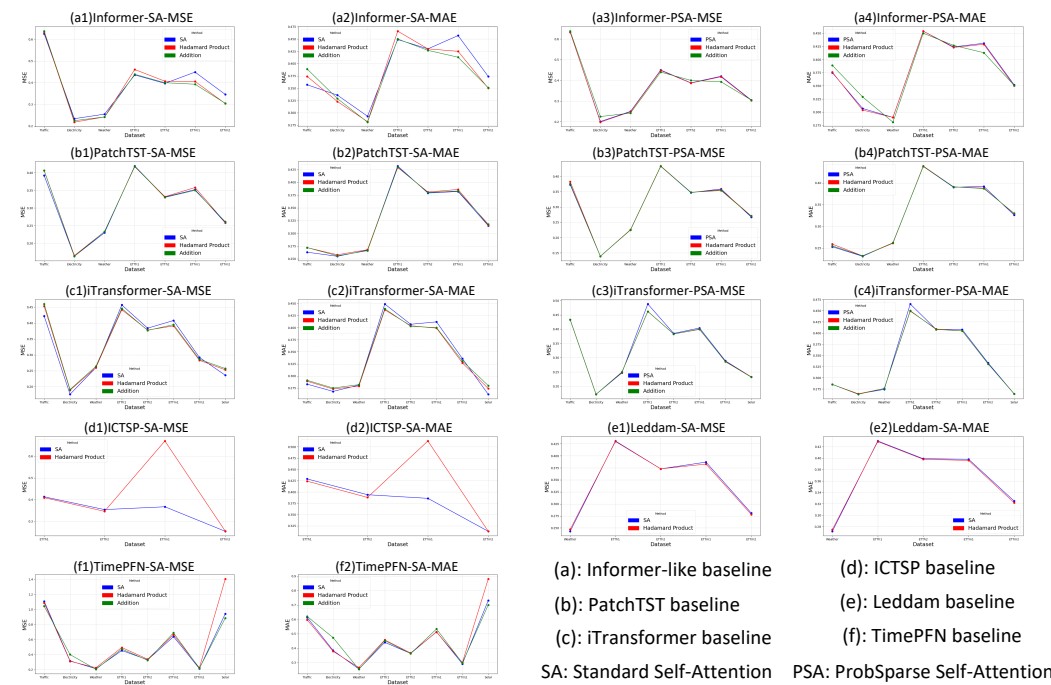

Figure 3: Visualization display of the ablation study. From a global perspective, the lines of the baseline and the ablation scheme are close together, indicating that removing the dynamic attention matrix has no significant impact. The figure shows the average error for 4 prediction horizons, the complete results of baseline (a) to (f) are in the Appendix from Table 3 to Table 8, respectively.

calculate the win rate, which represents the proportion of cases where the error of the ablation scheme is less than or equal to that of the baseline.

**Analysis on the time-token paradigm.** We first compare two simple element-wise operations against the Standard Self-Attention (SA). As shown in Figures 3 (a1) and (a2), in terms of overall average performance, the Hadamard version leads to a 2.59% decrease in MSE and a 1.71% decrease in MAE. The addition version yields even greater improvements, with a 3.68% decrease in MSE and a 2.08% decrease in MAE. Across all the 70 individual test cases with 4 prediction horizons, the Hadamard ablation performs better in 44 cases (62.86%), while the addition ablation wins in 47 cases (67.14%).

A similar trend is observed when ablating the more efficient ProbSparse Self-Attention (PSA), as shown in Figures 3 (a3) and (a4). The Hadamard version is superior in 58 out of 70 cases (82.86%) and also slightly improves the overall average performance. The addition version, while winning in just over half the cases (39/70, 55.71%), maintains a performance level highly comparable to the PSA baseline, with only a negligible 0.27% increase in MAE. Detailed quantitative results with 4 prediction horizons are in Table 3 of the Appendix.

For the Informer-like model, both dense and sparse matrix multiplications can be effectively replicated and often surpassed by simpler element-wise operations.

**Analysis on patch-token paradigm.** Compared to SA, as shown in Figures 3 (b1) and (b2), we observe a high degree of performance preservation. The impact on overall average performance is minimal: the Hadamard version shows a mere 1.16% increase in MSE and a 0.69% increase in MAE, while the addition version shows a 0.81% increase in MSE and a 0.54% increase in MAE. This marginal performance change (<1.2%) indicates that removing the matrix multiplication module, widely considered the soul of SA, has almost no substantive impact on the predictive power of the model.

This phenomenon is even more pronounced against the PSA baseline, as shown in Figures 3 (b3) and (b4). Across 58 valid test cases with 4 prediction horizons, while the win rates for the simpli-

fied versions are not high, the overall average performance shows 0.00% change in MAE for both ablations, with MSE fluctuations contained within ±0.2%. This demonstrates that replacing PSA with simple element-wise operations preserves performance almost perfectly. Detailed quantitative results with 4 prediction horizons are in Table 4 of the Appendix.

PatchTST results provide compelling evidence that complex matrix multiplication, standard or sparse, is not the source of its strong performance. Performance is almost entirely maintained after removing this core computational block.

**Analysis on variate-token paradigm.** Against SA, Hadamard and addition ablations achieve win rates of 62.50% and 52.50% across all 80 individual test cases with 4 prediction horizons, respectively. Interestingly, despite the high win rates, overall average performance is very close to the SA baseline, with error fluctuations contained within 1.6%, as shown in Figures 3 (c1) and (c2). Against PSA, we observe an overwhelming advantage for the simplified methods: the win rates for Hadamard and addition soar to 77.50% and 83.75% across all 80 individual test cases with 4 prediction horizons, respectively. Their overall average performance is also superior to the PSA baseline, with varying degrees of reduction in both MSE and MAE, as shown in Figures 3 (c3) and (c4). Detailed quantitative results with 4 prediction horizons are in Table 5 of the Appendix.

The iTransformer results reinforce our core thesis. Simple operations demonstrate comparable or superior performance in terms of both win rate and average metrics, indicating that complex attention calculations are equally unnecessary in the variate-dimension attention paradigm.

**Analysis on mixed paradigm 'ICTSP'.** To test the robustness of our findings on the latest generation of models, we first examine ICTSP, a representative hybrid paradigm model. The results present a more complex but ultimately supportive picture.

At first glance, the win rate suggests our thesis holds: across 40 valid test cases with 4 prediction horizons, the Hadamard ablation achieves a lower or equal error in 24 cases (60.00%). This demonstrates that even in this modern architecture, the simple element-wise operation is competitive or superior in a majority of scenarios. However, the overall average performance tells a different story, with the ablation's MSE significantly increasing by 21.04% and MAE increasing by 7.62%, as shown in Figures 3 (d1) and (d2). A deeper look reveals this is due to a 'catastrophic failure' on the ETTm1 dataset's long-horizon tasks, where the error amplified dramatically. On other datasets like ETTh1, the Hadamard version was clearly superior. This suggests that while our core thesis that matrix multiplication is not the primary performance driver holds, its role may shift to that of a stabilizer in certain contexts. Detailed quantitative results with 4 prediction horizons are in Table 6 of the Appendix.

**Analysis on mixed paradigm 'Leddam'.** Continuing our investigation into hybrid paradigms, we conduct ablation studies on the Leddam model. The results from this model provide a strong counterpoint to the robustness concerns raised by ICTSP and reinforce our thesis.

Across all 54 valid test cases with 4 prediction horizons, the Hadamard ablation achieves a lower or equal error in 38 cases, a high 70.37% win rate. This decisive majority indicates a clear advantage for the simpler mechanism within this architecture. Unlike with ICTSP, this superiority is also reflected in overall average performance, as shown in Figures 3 (e1) and (e2). The simplified version is comparable to the baseline, with MSE and MAE improving by a slight 0.23% and 0.22%, respectively. This confirms that our findings remain valid on the latest models, proving that matrix multiplication is not the key performance driver. Detailed quantitative results with 4 prediction horizons are in Table 7 of the Appendix.

**Analysis on mixed paradigm 'TimePFN'.** Since TimePFN only has open-sourced pretrained checkpoints instead of full training scripts, all our ablation experiments are conducted in inference mode. This means our element-wise operation modules are not trained at all but directly replace the corresponding parts in the original, fully trained model. This creates a stringent test of whether the architectural structure alone is sufficient.

As shown in Figure 3 (f1) and (f2), across 16 test cases, the Hadamard ablation wins in 5 cases (31.25%), while the addition version wins in 7 cases (43.75%). Detailed quantitative results are in Table 8 of the Appendix. While these win rates are below 50%, they must be interpreted in the context of the experiment: these simple, untrained operations are competing with a fully trained, complex module. The fact that they can remain competitive and even win in a significant minority of

cases is a highly unusual and compelling result. It suggests that the learned parameters of the matrix multiplication are less important than the simple act of fusing the Q and K branches. Therefore, it provides one of the strongest pieces of evidence for our core thesis.

## 4.2 RQ2: DOES THE PERFORMANCE OF SELF-ATTENTION PRIMARILY STEM FROM MULTI-BRANCH FEATURE MAPPING AND FUSION?

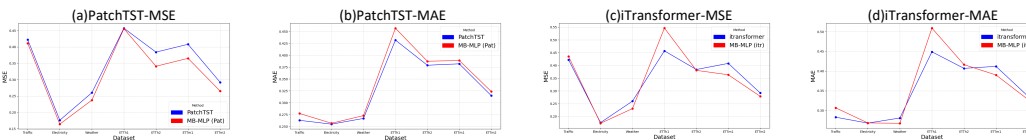

Figure 4: Visualization display of the comparison between MB-MLP and two well-known essential paradigms. From a global perspective, the lines of the baseline and the MB-MLP are very close, which confirms our hypothesis. The figure shows the average error for 4 prediction horizons, the complete results with 4 prediction horizons are in the Appendix Table 9.

Our findings in RQ1 indicate that the dynamic attention matrix calculation is not the key to performance. This naturally leads to the next question: if the performance does not originate from the attention calculation, where does it come from? We hypothesize that the foundation of the performance primarily stems from the long-overlooked multi-branch feature mapping and simple fusion architecture that precedes the attention calculation. Other components, we posit, provide auxiliary performance gains on top of this foundation.

To validate this core hypothesis, we construct a minimalist Multi-Branch MLP (MB-MLP) model, as illustrated in Figure 2. This model completely strips away the complex components of a Transformer, such as FFN, retaining only what we consider the core structure: 'QKV multi-branch mapping + simple fusion'. We test the performance of the MB-MLP according to the 3 essential tokenization paradigms, comparing it against the corresponding full Transformer models. As long as the MB-MLP can approach, without having to surpass, the performance of the baselines, our hypothesis can be confirmed.

**Validation on the patch-token paradigm.** On the patch-token paradigm, MB-MLP (Pat) (red line), as shown in Figure 4 (a) and (b), successfully reproduces the main body of PatchTST's accuracy. It should be emphasized that the performance of MB-MLP (Pat) only needs to be close to, rather than surpass, the baseline to verify our hypothesis. Specifically, the overall average MSE and MAE for the MB-MLP increase by 4.38% and 3.12%, respectively. However, this result must be interpreted in the context of the vast difference in model complexity. The MB-MLP removes all attention calculations, layer normalizations, and residual connections from PatchTST, resulting in a significant simplification of its overhead (time and memory are reduced by about 89% and 88% respectively) and computational flow. In this context, a performance gap of only 3-4% is strong evidence for our core hypothesis: the QKV multi-branch mapping structure successfully captures the vast majority of the PatchTST model's predictive power. In other words, a multi-branch feature fusion MLP with minimal overhead is already capable of achieving over 95% of the Transformer's core accuracy. Complete quantitative results with 4 prediction horizons are in the Appendix Table 9.

**Validation on the variate-token paradigm.** Figure 4 (c) and (d) prove that MB-MLP (iTr) can reproduce the main body of iTransformer's accuracy at a minimal performance cost with an input length of 336. We observe that the overall average MSE and MAE for the MB-MLP increase by 0.38% and 2.07%, respectively. Its performance is almost on par with the baseline, which fully verifies our hypothesis. Complete quantitative results with 4 prediction horizons are in the Appendix Table 9.

**Validation on the time-token paradigm.** MB-MLP (Inf) achieves a performance highly comparable to the full Informer model. In terms of overall average performance, the two are remarkably close: MB-MLP reduces 1.57% MSE and 0.37% MAE. This high degree of consistency in overall performance, combined with its wins in a majority of individual cases, clearly demonstrates that removing most of the Transformer's complex components does not substantially impact the model's

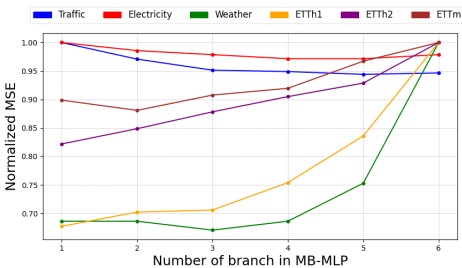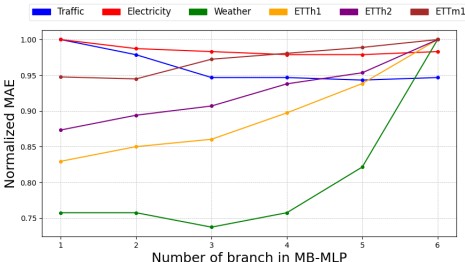

Figure 5: This ablation study focuses on the number of branches, with the purpose of validating the hypothesis rather than comparing performance. The substantial global fluctuations provide evidence that multi-branch feature mapping and fusion play a central role.

core predictive capability. Complete quantitative results with 4 prediction horizons are in the Appendix Table 9.

**Ablation Study on Branches Number.** To further validate the hypothesis, we conduct an ablation study on the number of branches in MB-MLP, ranging from 1 to 6. The specific implementation details are in the Appendix A.3. If a component were unimportant, varying it would not lead to substantial performance changes. However, Figure 5 shows that all metrics across datasets fluctuate markedly with the number of branches, strongly confirming that multi-branch feature mapping and fusion are the key factors, thereby supporting the hypothesis.

### 4.3 RQ3: Is the attention matrix worth the computational cost?

Table 2: Overhead comparison between attention matrix and Hadamard product matrix based on 3 essential paradigms on the largest-scaled dataset Traffic. Red represents the best results. The attention matrix incurs higher computational overhead but yields only marginal performance gains.

| Horizon | Metric / Model | Informer | Hadamard | PatchTST | Hadamard | iTransformer | Hadamard |
|---------|---------------|----------|----------|----------|----------|--------------|----------|
| 96 | Time (s/epoch) | **44.172** | 45.621 | 125.640 | **63.862** | 50.800 | **26.757** |
| | Memory (GB) | 3.328 | **2.408** | 36.576 | **26.240** | 6.064 | **2.992** |
| 192 | Time (s/epoch) | **47.422** | 48.339 | 127.352 | **65.549** | 51.920 | **27.799** |
| | Memory (GB) | 3.364 | **2.440** | 36.582 | **26.460** | 6.072 | **3.022** |
| 336 | Time (s/epoch) | **52.812** | 58.194 | 130.900 | **68.553** | 53.136 | **29.825** |
| | Memory (GB) | 3.438 | **2.592** | 36.614 | **26.492** | 6.456 | **3.116** |
| 720 | Time (s/epoch) | 62.845 | **61.755** | 139.331 | **78.196** | 58.620 | **36.067** |
| | Memory (GB) | 3.760 | **3.088** | 37.224 | **26.544** | 6.504 | **3.380** |

Our findings from RQ1 and RQ2 raise a practical question: is it worthwhile to retain the non-essential yet computationally expensive attention matrix? The answer, based on our analysis, is no. As shown in Table 2, the attention matrix reduces average training time by only 3.11% compared to the Hadamard matrix in the Informer-like paradigm, but incurs higher overhead in all other cases. Specifically, across the three paradigms from left to right, the attention matrix is 3.11% faster and consumes 31.93% more memory, 89.46% slower and 39.02% more memory, and 78.07% slower and 100.61% more memory, respectively.

## 5 CONCLUSION

This paper systematically investigates the true role of attention matrix in multivariate long-term time series forecasting through extensive ablations on 6 classic and latest Transformer-based models. Our results consistently show that the dynamic attention matrix is significantly overestimated and can be replaced by simple element-wise operations without significant performance loss. We validate our core hypothesis with a minimalist multi-branch MLP, proving the performance foundation is the front-end QKV multi-branch mapping and fusion architecture. These findings suggest the community has misunderstood the source of success in attention models, opening a new direction for simpler and more efficient forecasting architectures.

## 6 REPRODUCIBILITY STATEMENT

For novel models or algorithms, a link to an anonymous source code is provided at the abstract section; For the datasets used in the experiments, a detailed description is provided in the Appendix A.2.

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

## A APPENDIX

### A.1 THE USE OF LARGE LANGUAGE MODELS

We employed a Large Language Model as a general-purpose writing assistant. Specifically, it was used for language refinement, content polishing, and occasionally for suggesting alternative phrasings to improve readability and clarity.

### A.2 DATASETS

Following baselines, we employ their real-world benchmark datasets for evaluation. All these datasets can be obtained from Wu et al. (2021); Liu et al. (2023); Wang et al. (2024b). According to FreTS Yi et al. (2024), the details of these datasets are as follows:

**Traffic**[1]: This dataset records hourly traffic flow data for 963 freeway lanes in San Francisco. It supports long-term forecasting using 862 lanes. The data collection began on January 1, 2015, with an hourly sampling interval.

**Electricity**[2]: This dataset captures electricity consumption patterns of 370 clients for short-term forecasting and 321 clients for long-term predictions. The data spans from January 1, 2011, with a 15-minute sampling interval.

**Solar**[3]: This dataset consists of solar power measurements collected by the National Renewable Energy Laboratory. It includes data points from various power plants in Florida, comprising 593 records, spanning from January 1, 2006, to December 31, 2016, with an hourly sampling interval.

---

[1] https://pems.dot.ca.gov/
[2] https://archive.ics.uci.edu/ml/datasets/ElectricityLoadDiagrams20112014
[3] https://www.nrel.gov/grid/solar-power-data.html

**ETT**[4]: This dataset includes four subsets (ETTh1, ETTh2, ETTm1, and ETTm2). It measurements from two distinct electric transformers, labeled ETTh1 and ETTm1, representing different temporal resolutions (hourly and 15-minute intervals, respectively). These datasets serve as benchmarks for long-term forecasting.

### A.3    Implementation Details

For the 6 baselines, our ablation experiments fully follow the implementation details of their open-source code, including hyperparameters and so on. The baseline 'Informer-like' is constructed by following the original paradigm, with the aim of excluding potential effects introduced by the ineffective decoder component in Informer.

In Figure 1 and 2, $D$ is embedding dimension.

In the MB-MLP (iTr) of RQ2, the input sequence length is set to 336 because 96, which is set by its baseline, i.e., iTransformer, is not applicable. Specifically, we find that the MB-MLP (iTr) exhibits a positive sensitivity to the input sequence length. In preliminary experiments, the model performed poorly with a shorter input length of 96. However, when we increased the input length to 336, its performance improved significantly. This phenomenon suggests that our minimalist MB-MLP model is not only effective but that its multi-branch feature mapping structure can efficiently encode information from longer input sequences to serve long-term forecasting.

To investigate the optimal number of mapping branches applicable to varying scenarios, we performed an ablation study on the MB-MLP model under the PatchTST framework. Specifically, the number of branches was adjusted from 1 to 6, with evaluations conducted across 7 datasets under a 96-step prediction horizon. Nonlinear activation function is removed to construct a model that tests only the hypothesized conditions, excluding other sensitive factors.

### A.4    Results and Analysis

---

[4]https://github.com/zhouhaoyi/ETDataset

Table 3: Ablation results of the time-token paradigm, where 'SA', 'Hadam Prod', 'Addition', and 'PSA' represent the self-attention, Hadamard product, addition, and sparse self-attention, respectively. Every three adjacent ones form a group of ablation experiments. The Hadamard product and addition have lower MSE and MAE than the attention mechanisms in more cases.

| Models | | Informer-like | | | | | | | | | | | |
|---|---|---|---|---|---|---|---|---|---|---|---|---|---|
| | | SA | | Hadam Prod | | Addition | | PSA | | Hadam Prod | | Addition | |
| Metric | | MSE | MAE | MSE | MAE | MSE | MAE | MSE | MAE | MSE | MAE | MSE | MAE |
| Traffic | 96 | 0.614 | 0.350 | 0.634 | 0.372 | 0.648 | 0.390 | 0.622 | 0.360 | 0.627 | 0.363 | 0.648 | 0.390 |
| | 192 | 0.618 | 0.352 | 0.605 | 0.370 | 0.601 | 0.386 | 0.615 | 0.382 | 0.611 | 0.381 | 0.601 | 0.386 |
| | 336 | 0.637 | 0.358 | 0.639 | 0.377 | 0.640 | 0.385 | 0.624 | 0.373 | 0.624 | 0.374 | 0.640 | 0.385 |
| | 720 | 0.636 | 0.367 | 0.658 | 0.377 | 0.663 | 0.393 | 0.671 | 0.385 | 0.672 | 0.386 | 0.663 | 0.393 |
| | AVG | 0.626 | 0.357 | 0.634 | 0.374 | 0.638 | 0.389 | 0.633 | 0.375 | 0.634 | 0.376 | 0.638 | 0.389 |
| Electricity | 96 | 0.214 | 0.319 | 0.210 | 0.314 | 0.209 | 0.313 | 0.181 | 0.288 | 0.181 | 0.288 | 0.209 | 0.313 |
| | 192 | 0.273 | 0.369 | 0.208 | 0.314 | 0.220 | 0.323 | 0.205 | 0.310 | 0.201 | 0.306 | 0.220 | 0.323 |
| | 336 | 0.228 | 0.332 | 0.224 | 0.332 | 0.229 | 0.335 | 0.199 | 0.309 | 0.197 | 0.307 | 0.229 | 0.335 |
| | 720 | 0.220 | 0.323 | 0.228 | 0.333 | 0.241 | 0.344 | 0.217 | 0.319 | 0.213 | 0.316 | 0.241 | 0.344 |
| | AVG | 0.234 | 0.336 | 0.218 | 0.323 | 0.225 | 0.329 | 0.201 | 0.307 | 0.198 | 0.304 | 0.225 | 0.329 |
| Weather | 96 | 0.180 | 0.238 | 0.170 | 0.225 | 0.168 | 0.223 | 0.169 | 0.229 | 0.168 | 0.229 | 0.168 | 0.223 |
| | 192 | 0.217 | 0.268 | 0.212 | 0.261 | 0.211 | 0.260 | 0.212 | 0.266 | 0.211 | 0.265 | 0.211 | 0.260 |
| | 336 | 0.270 | 0.305 | 0.263 | 0.298 | 0.263 | 0.299 | 0.265 | 0.304 | 0.263 | 0.303 | 0.263 | 0.299 |
| | 720 | 0.351 | 0.362 | 0.324 | 0.342 | 0.327 | 0.343 | 0.354 | 0.362 | 0.351 | 0.361 | 0.327 | 0.343 |
| | AVG | 0.255 | 0.293 | 0.242 | 0.282 | 0.242 | 0.281 | 0.250 | 0.290 | 0.248 | 0.290 | 0.242 | 0.281 |
| ETTh1 | 96 | 0.395 | 0.421 | 0.473 | 0.476 | 0.394 | 0.420 | 0.410 | 0.426 | 0.410 | 0.426 | 0.394 | 0.420 |
| | 192 | 0.423 | 0.437 | 0.428 | 0.439 | 0.430 | 0.439 | 0.430 | 0.434 | 0.430 | 0.434 | 0.430 | 0.439 |
| | 336 | 0.444 | 0.451 | 0.446 | 0.449 | 0.442 | 0.446 | 0.456 | 0.459 | 0.457 | 0.459 | 0.443 | 0.446 |
| | 720 | 0.483 | 0.487 | 0.497 | 0.498 | 0.491 | 0.493 | 0.505 | 0.498 | 0.494 | 0.495 | 0.491 | 0.493 |
| | AVG | 0.436 | 0.449 | 0.461 | 0.466 | 0.439 | 0.450 | 0.450 | 0.454 | 0.448 | 0.454 | 0.440 | 0.450 |
| ETTh2 | 96 | 0.382 | 0.412 | 0.383 | 0.413 | 0.377 | 0.407 | 0.361 | 0.395 | 0.361 | 0.394 | 0.377 | 0.407 |
| | 192 | 0.405 | 0.435 | 0.427 | 0.433 | 0.407 | 0.424 | 0.392 | 0.431 | 0.391 | 0.430 | 0.407 | 0.424 |
| | 336 | 0.380 | 0.425 | 0.377 | 0.421 | 0.372 | 0.420 | 0.359 | 0.412 | 0.358 | 0.411 | 0.371 | 0.420 |
| | 720 | 0.419 | 0.446 | 0.439 | 0.454 | 0.445 | 0.456 | 0.439 | 0.457 | 0.437 | 0.455 | 0.445 | 0.456 |
| | AVG | 0.397 | 0.430 | 0.407 | 0.430 | 0.400 | 0.427 | 0.388 | 0.424 | 0.387 | 0.423 | 0.400 | 0.427 |
| ETTm1 | 96 | 0.399 | 0.425 | 0.323 | 0.374 | 0.319 | 0.368 | 0.349 | 0.387 | 0.339 | 0.381 | 0.319 | 0.368 |
| | 192 | 0.422 | 0.439 | 0.372 | 0.402 | 0.363 | 0.394 | 0.403 | 0.418 | 0.388 | 0.409 | 0.363 | 0.394 |
| | 336 | 0.448 | 0.455 | 0.404 | 0.421 | 0.401 | 0.417 | 0.417 | 0.428 | 0.439 | 0.441 | 0.401 | 0.417 |
| | 720 | 0.526 | 0.509 | 0.526 | 0.504 | 0.489 | 0.472 | 0.510 | 0.490 | 0.502 | 0.485 | 0.489 | 0.472 |
| | AVG | 0.449 | 0.457 | 0.406 | 0.425 | 0.393 | 0.413 | 0.420 | 0.431 | 0.417 | 0.429 | 0.393 | 0.413 |
| ETTm2 | 96 | 0.239 | 0.307 | 0.220 | 0.298 | 0.217 | 0.298 | 0.199 | 0.283 | 0.196 | 0.280 | 0.217 | 0.298 |
| | 192 | 0.319 | 0.354 | 0.259 | 0.324 | 0.261 | 0.322 | 0.277 | 0.333 | 0.276 | 0.334 | 0.261 | 0.322 |
| | 336 | 0.341 | 0.380 | 0.322 | 0.361 | 0.324 | 0.364 | 0.331 | 0.374 | 0.327 | 0.370 | 0.324 | 0.364 |
| | 720 | 0.485 | 0.453 | 0.415 | 0.416 | 0.419 | 0.419 | 0.413 | 0.417 | 0.414 | 0.417 | 0.419 | 0.419 |
| | AVG | 0.346 | 0.374 | 0.304 | 0.350 | 0.305 | 0.351 | 0.305 | 0.352 | 0.303 | 0.350 | 0.305 | 0.351 |

Table 4: Ablation results of the patch-token paradigm. Hadamard product and addition can maintain the accuracy of the attention mechanism. Due to the excessive computational overhead (over 400 s/epoch) of long-sequence forecasting on the large-scale Traffic and Electricity datasets, we were unable to obtain results for all prediction horizons.

| Models | | PatchTST | | | | | | | | | | | |
|--------|------|-----|-----|------------|-----|----------|-----|-----|-----|------------|-----|----------|-----|
| | | SA | | Hadam Prod | | Addition | | PSA | | Hadam Prod | | Addition | |
| Metric | | MSE | MAE | MSE | MAE | MSE | MAE | MSE | MAE | MSE | MAE | MSE | MAE |
| Traffic | 96 | 0.361 | 0.247 | 0.382 | 0.261 | 0.381 | 0.259 | 0.364 | 0.249 | 0.374 | 0.256 | 0.366 | 0.250 |
| | 192 | 0.380 | 0.255 | 0.394 | 0.265 | 0.395 | 0.265 | 0.382 | 0.257 | 0.391 | 0.262 | 0.386 | 0.259 |
| | 336 | 0.393 | 0.263 | 0.406 | 0.271 | 0.407 | 0.271 | - | - | - | - | - | - |
| | 720 | 0.432 | 0.286 | 0.440 | 0.291 | 0.441 | 0.292 | - | - | - | - | - | - |
| | AVG | 0.392 | 0.263 | 0.406 | 0.272 | 0.406 | 0.272 | - | - | - | - | - | - |
| Electricity | 96 | 0.130 | 0.222 | 0.134 | 0.228 | 0.134 | 0.228 | 0.130 | 0.223 | 0.130 | 0.224 | 0.130 | 0.224 |
| | 192 | 0.148 | 0.240 | 0.149 | 0.242 | 0.149 | 0.243 | 0.147 | 0.239 | 0.147 | 0.240 | 0.147 | 0.240 |
| | 336 | 0.165 | 0.259 | 0.166 | 0.261 | 0.165 | 0.260 | 0.164 | 0.258 | - | - | - | - |
| | 720 | 0.210 | 0.298 | 0.210 | 0.299 | 0.204 | 0.294 | 0.200 | 0.289 | - | - | - | - |
| | AVG | 0.163 | 0.255 | 0.165 | 0.258 | 0.163 | 0.256 | 0.160 | 0.252 | - | - | - | - |
| Weather | 96 | 0.156 | 0.205 | 0.157 | 0.206 | 0.157 | 0.204 | 0.149 | 0.197 | 0.149 | 0.196 | 0.148 | 0.195 |
| | 192 | 0.195 | 0.242 | 0.201 | 0.246 | 0.202 | 0.244 | 0.192 | 0.238 | 0.192 | 0.239 | 0.191 | 0.239 |
| | 336 | 0.250 | 0.284 | 0.250 | 0.284 | 0.251 | 0.283 | 0.244 | 0.278 | 0.244 | 0.279 | 0.243 | 0.280 |
| | 720 | 0.320 | 0.335 | 0.325 | 0.336 | 0.320 | 0.333 | 0.319 | 0.334 | 0.317 | 0.331 | 0.317 | 0.332 |
| | AVG | 0.230 | 0.267 | 0.233 | 0.268 | 0.233 | 0.266 | 0.226 | 0.262 | 0.226 | 0.261 | 0.225 | 0.262 |
| ETTh1 | 96 | 0.382 | 0.405 | 0.384 | 0.407 | 0.385 | 0.409 | 0.391 | 0.408 | 0.393 | 0.409 | 0.391 | 0.408 |
| | 192 | 0.414 | 0.421 | 0.414 | 0.421 | 0.415 | 0.422 | 0.423 | 0.424 | 0.424 | 0.425 | 0.423 | 0.423 |
| | 336 | 0.431 | 0.435 | 0.427 | 0.429 | 0.427 | 0.430 | 0.433 | 0.434 | 0.433 | 0.435 | 0.433 | 0.434 |
| | 720 | 0.449 | 0.466 | 0.441 | 0.457 | 0.445 | 0.460 | 0.483 | 0.484 | 0.484 | 0.485 | 0.484 | 0.485 |
| | AVG | 0.419 | 0.432 | 0.417 | 0.429 | 0.418 | 0.430 | 0.433 | 0.438 | 0.434 | 0.439 | 0.433 | 0.438 |
| ETTh2 | 96 | 0.274 | 0.336 | 0.276 | 0.337 | 0.275 | 0.336 | 0.291 | 0.345 | 0.291 | 0.345 | 0.292 | 0.346 |
| | 192 | 0.339 | 0.379 | 0.339 | 0.379 | 0.338 | 0.379 | 0.358 | 0.389 | 0.360 | 0.391 | 0.359 | 0.389 |
| | 336 | 0.331 | 0.381 | 0.331 | 0.385 | 0.329 | 0.384 | 0.342 | 0.389 | 0.343 | 0.390 | 0.343 | 0.390 |
| | 720 | 0.379 | 0.421 | 0.381 | 0.423 | 0.378 | 0.421 | 0.398 | 0.437 | 0.398 | 0.437 | 0.399 | 0.437 |
| | AVG | 0.331 | 0.379 | 0.332 | 0.381 | 0.330 | 0.380 | 0.347 | 0.390 | 0.348 | 0.391 | 0.348 | 0.391 |
| ETTm1 | 96 | 0.292 | 0.343 | 0.304 | 0.350 | 0.292 | 0.347 | 0.293 | 0.347 | 0.293 | 0.345 | 0.288 | 0.342 |
| | 192 | 0.331 | 0.369 | 0.344 | 0.377 | 0.332 | 0.371 | 0.337 | 0.378 | 0.335 | 0.374 | 0.333 | 0.373 |
| | 336 | 0.365 | 0.392 | 0.368 | 0.394 | 0.365 | 0.393 | 0.374 | 0.404 | 0.366 | 0.397 | 0.363 | 0.393 |
| | 720 | 0.419 | 0.425 | 0.415 | 0.424 | 0.410 | 0.420 | 0.430 | 0.438 | 0.429 | 0.437 | 0.431 | 0.439 |
| | AVG | 0.352 | 0.382 | 0.358 | 0.386 | 0.350 | 0.383 | 0.359 | 0.392 | 0.356 | 0.388 | 0.354 | 0.387 |
| ETTm2 | 96 | 0.165 | 0.255 | 0.164 | 0.253 | 0.164 | 0.254 | 0.172 | 0.262 | 0.175 | 0.264 | 0.174 | 0.265 |
| | 192 | 0.220 | 0.292 | 0.222 | 0.293 | 0.222 | 0.293 | 0.233 | 0.305 | 0.238 | 0.304 | 0.234 | 0.306 |
| | 336 | 0.278 | 0.329 | 0.279 | 0.330 | 0.279 | 0.330 | 0.285 | 0.336 | 0.292 | 0.347 | 0.294 | 0.348 |
| | 720 | 0.367 | 0.385 | 0.372 | 0.389 | 0.377 | 0.396 | 0.377 | 0.402 | 0.377 | 0.401 | 0.378 | 0.402 |
| | AVG | 0.258 | 0.315 | 0.259 | 0.316 | 0.261 | 0.318 | 0.267 | 0.326 | 0.271 | 0.329 | 0.270 | 0.330 |

Table 5: Ablation results of the variate-token paradigm. The Hadamard product and addition have lower MSE and MAE than the attention mechanisms in more cases.

| Models | | iTransformer | | | | | | | | | | | |
|---|---|---|---|---|---|---|---|---|---|---|---|---|---|
| | | SA | | Hadam Prod | | Addition | | PSA | | Hadam Prod | | Addition | |
| Metric | | MSE | MAE | MSE | MAE | MSE | MAE | MSE | MAE | MSE | MAE | MSE | MAE |
| Traffic | 96 | 0.392 | 0.269 | 0.428 | 0.277 | 0.436 | 0.280 | 0.400 | 0.270 | 0.400 | 0.270 | 0.400 | 0.270 |
| | 192 | 0.414 | 0.278 | 0.441 | 0.282 | 0.448 | 0.284 | 0.422 | 0.279 | 0.422 | 0.279 | 0.422 | 0.279 |
| | 336 | 0.424 | 0.283 | 0.455 | 0.288 | 0.462 | 0.291 | 0.438 | 0.286 | 0.438 | 0.286 | 0.438 | 0.286 |
| | 720 | 0.458 | 0.300 | 0.488 | 0.308 | 0.494 | 0.310 | 0.470 | 0.304 | 0.470 | 0.304 | 0.470 | 0.304 |
| | AVG | 0.422 | 0.283 | 0.453 | 0.289 | 0.460 | 0.291 | 0.433 | 0.285 | 0.433 | 0.285 | 0.433 | 0.285 |
| Electricity | 96 | 0.148 | 0.240 | 0.164 | 0.249 | 0.166 | 0.251 | 0.144 | 0.237 | 0.144 | 0.237 | 0.144 | 0.237 |
| | 192 | 0.168 | 0.259 | 0.173 | 0.258 | 0.175 | 0.260 | 0.159 | 0.250 | 0.160 | 0.250 | 0.160 | 0.250 |
| | 336 | 0.178 | 0.271 | 0.189 | 0.275 | 0.192 | 0.277 | 0.174 | 0.267 | 0.175 | 0.267 | 0.175 | 0.266 |
| | 720 | 0.211 | 0.301 | 0.228 | 0.309 | 0.232 | 0.311 | 0.210 | 0.303 | 0.208 | 0.300 | 0.208 | 0.300 |
| | AVG | 0.176 | 0.268 | 0.189 | 0.273 | 0.191 | 0.275 | 0.172 | 0.264 | 0.172 | 0.264 | 0.172 | 0.263 |
| Weather | 96 | 0.176 | 0.216 | 0.178 | 0.216 | 0.181 | 0.220 | 0.159 | 0.204 | 0.163 | 0.208 | 0.163 | 0.207 |
| | 192 | 0.225 | 0.257 | 0.225 | 0.257 | 0.229 | 0.260 | 0.209 | 0.249 | 0.212 | 0.252 | 0.212 | 0.252 |
| | 336 | 0.281 | 0.299 | 0.280 | 0.296 | 0.284 | 0.299 | 0.268 | 0.293 | 0.269 | 0.294 | 0.271 | 0.295 |
| | 720 | 0.358 | 0.350 | 0.357 | 0.346 | 0.361 | 0.350 | 0.352 | 0.348 | 0.350 | 0.349 | 0.353 | 0.350 |
| | AVG | 0.260 | 0.281 | 0.260 | 0.279 | 0.264 | 0.282 | 0.247 | 0.274 | 0.249 | 0.276 | 0.250 | 0.276 |
| ETTh1 | 96 | 0.387 | 0.405 | 0.381 | 0.397 | 0.381 | 0.397 | 0.391 | 0.407 | 0.387 | 0.403 | 0.386 | 0.403 |
| | 192 | 0.441 | 0.436 | 0.430 | 0.426 | 0.433 | 0.427 | 0.446 | 0.438 | 0.441 | 0.433 | 0.441 | 0.434 |
| | 336 | 0.491 | 0.462 | 0.470 | 0.447 | 0.479 | 0.452 | 0.495 | 0.462 | 0.492 | 0.460 | 0.493 | 0.461 |
| | 720 | 0.509 | 0.494 | 0.487 | 0.479 | 0.490 | 0.481 | 0.620 | 0.554 | 0.526 | 0.501 | 0.529 | 0.502 |
| | AVG | 0.457 | 0.449 | 0.442 | 0.437 | 0.446 | 0.439 | 0.488 | 0.465 | 0.462 | 0.449 | 0.462 | 0.450 |
| ETTh2 | 96 | 0.301 | 0.350 | 0.295 | 0.347 | 0.293 | 0.345 | 0.297 | 0.348 | 0.294 | 0.347 | 0.294 | 0.346 |
| | 192 | 0.380 | 0.399 | 0.375 | 0.396 | 0.374 | 0.394 | 0.379 | 0.399 | 0.375 | 0.398 | 0.376 | 0.398 |
| | 336 | 0.424 | 0.432 | 0.417 | 0.430 | 0.417 | 0.430 | 0.424 | 0.435 | 0.423 | 0.436 | 0.423 | 0.435 |
| | 720 | 0.430 | 0.447 | 0.426 | 0.444 | 0.424 | 0.444 | 0.438 | 0.451 | 0.438 | 0.453 | 0.437 | 0.451 |
| | AVG | 0.384 | 0.407 | 0.378 | 0.404 | 0.377 | 0.403 | 0.385 | 0.408 | 0.383 | 0.409 | 0.383 | 0.408 |
| ETTm1 | 96 | 0.342 | 0.377 | 0.325 | 0.364 | 0.331 | 0.366 | 0.339 | 0.372 | 0.339 | 0.372 | 0.340 | 0.372 |
| | 192 | 0.383 | 0.396 | 0.369 | 0.385 | 0.373 | 0.385 | 0.375 | 0.390 | 0.372 | 0.388 | 0.372 | 0.389 |
| | 336 | 0.418 | 0.418 | 0.404 | 0.407 | 0.407 | 0.406 | 0.413 | 0.415 | 0.409 | 0.411 | 0.410 | 0.412 |
| | 720 | 0.487 | 0.457 | 0.467 | 0.441 | 0.469 | 0.441 | 0.489 | 0.456 | 0.476 | 0.448 | 0.476 | 0.448 |
| | AVG | 0.408 | 0.412 | 0.391 | 0.399 | 0.395 | 0.400 | 0.404 | 0.408 | 0.399 | 0.405 | 0.400 | 0.405 |
| ETTm2 | 96 | 0.186 | 0.272 | 0.178 | 0.260 | 0.181 | 0.266 | 0.181 | 0.266 | 0.179 | 0.264 | 0.179 | 0.264 |
| | 192 | 0.253 | 0.314 | 0.245 | 0.304 | 0.246 | 0.307 | 0.249 | 0.309 | 0.245 | 0.307 | 0.245 | 0.307 |
| | 336 | 0.316 | 0.351 | 0.306 | 0.344 | 0.309 | 0.347 | 0.310 | 0.349 | 0.311 | 0.349 | 0.310 | 0.349 |
| | 720 | 0.414 | 0.407 | 0.404 | 0.400 | 0.407 | 0.402 | 0.415 | 0.407 | 0.411 | 0.405 | 0.411 | 0.405 |
| | AVG | 0.292 | 0.336 | 0.283 | 0.327 | 0.286 | 0.331 | 0.289 | 0.333 | 0.287 | 0.331 | 0.286 | 0.331 |
| Solar | 96 | 0.205 | 0.236 | 0.221 | 0.254 | 0.226 | 0.261 | 0.199 | 0.237 | 0.199 | 0.238 | 0.198 | 0.237 |
| | 192 | 0.238 | 0.262 | 0.251 | 0.273 | 0.256 | 0.280 | 0.232 | 0.263 | 0.233 | 0.264 | 0.232 | 0.263 |
| | 336 | 0.250 | 0.274 | 0.271 | 0.285 | 0.275 | 0.291 | 0.248 | 0.276 | 0.248 | 0.276 | 0.248 | 0.276 |
| | 720 | 0.251 | 0.276 | 0.268 | 0.284 | 0.272 | 0.289 | 0.251 | 0.278 | 0.251 | 0.278 | 0.251 | 0.278 |
| | AVG | 0.236 | 0.262 | 0.253 | 0.274 | 0.257 | 0.280 | 0.233 | 0.264 | 0.233 | 0.264 | 0.232 | 0.264 |

Table 6: Ablation results of the latest model 'ICTSP' with mixed paradigm. The Hadamard product have lower MSE and MAE than the attention mechanisms in more cases. As its official implementation only provides training scripts for the ETT datasets, our experiments are confined to this scope.

| Models | | ETTh1 | | ETTh2 | | ETTm1 | | ETTm2 | |
|---|---|---|---|---|---|---|---|---|---|
| | | MSE | MAE | MSE | MAE | MSE | MAE | MSE | MAE |
| SA | 96 | 0.375 | 0.400 | 0.277 | 0.335 | 0.315 | 0.355 | 0.163 | 0.252 |
| | 192 | 0.404 | 0.415 | 0.340 | 0.376 | 0.352 | 0.380 | 0.219 | 0.291 |
| | 336 | 0.434 | 0.441 | 0.403 | 0.435 | 0.371 | 0.388 | 0.273 | 0.325 |
| | 720 | 0.437 | 0.460 | 0.396 | 0.431 | 0.431 | 0.421 | 0.361 | 0.382 |
| | AVG | 0.413 | 0.429 | 0.354 | 0.394 | 0.367 | 0.386 | 0.254 | 0.313 |
| Hadamard Product | 96 | 0.373 | 0.399 | 0.278 | 0.336 | 0.310 | 0.351 | 0.165 | 0.254 |
| | 192 | 0.406 | 0.417 | 0.332 | 0.376 | 0.503 | 0.455 | 0.219 | 0.291 |
| | 336 | 0.428 | 0.430 | 0.378 | 0.408 | 0.658 | 0.525 | 0.271 | 0.325 |
| | 720 | 0.430 | 0.449 | 0.396 | 0.431 | 1.210 | 0.721 | 0.363 | 0.382 |
| | AVG | 0.409 | 0.424 | 0.346 | 0.388 | 0.670 | 0.513 | 0.255 | 0.313 |

Table 7: Ablation results of the latest model 'Leddam' with mixed paradigm. The Hadamard product have lower MSE and MAE than the attention mechanisms in more cases. Due to the high cost of Leddam, on large-scale datasets Traffic and Electricity, we only measured a 96-step prediction horizon.

| Models | | Traffic | | Electricity | | Weather | | ETTh1 | | ETTh2 | | ETTm1 | | ETTm2 | |
|---|---|---|---|---|---|---|---|---|---|---|---|---|---|---|---|
| | | MSE | MAE | MSE | MAE | MSE | MAE | MSE | MAE | MSE | MAE | MSE | MAE | MSE | MAE |
| SA | 96 | 0.440 | 0.283 | 0.141 | 0.234 | 0.157 | 0.202 | 0.377 | 0.394 | 0.292 | 0.343 | 0.319 | 0.359 | 0.176 | 0.257 |
| | 192 | - | - | - | - | 0.207 | 0.249 | 0.424 | 0.422 | 0.367 | 0.389 | 0.371 | 0.384 | 0.243 | 0.303 |
| | 336 | - | - | - | - | 0.265 | 0.293 | 0.459 | 0.442 | 0.412 | 0.424 | 0.398 | 0.404 | 0.303 | 0.341 |
| | 720 | - | - | - | - | 0.343 | 0.343 | 0.464 | 0.460 | 0.419 | 0.438 | 0.461 | 0.443 | 0.400 | 0.398 |
| | AVG | - | - | - | - | 0.243 | 0.272 | 0.431 | 0.430 | 0.373 | 0.399 | 0.387 | 0.398 | 0.281 | 0.325 |
| Hadmard Product | 96 | 0.466 | 0.278 | 0.150 | 0.240 | 0.164 | 0.209 | 0.378 | 0.393 | 0.291 | 0.340 | 0.317 | 0.356 | 0.175 | 0.256 |
| | 192 | - | - | - | - | 0.210 | 0.251 | 0.423 | 0.422 | 0.368 | 0.389 | 0.362 | 0.380 | 0.240 | 0.301 |
| | 336 | - | - | - | - | 0.266 | 0.292 | 0.457 | 0.441 | 0.414 | 0.425 | 0.393 | 0.403 | 0.302 | 0.338 |
| | 720 | - | - | - | - | 0.347 | 0.346 | 0.462 | 0.460 | 0.419 | 0.437 | 0.458 | 0.443 | 0.395 | 0.394 |
| | AVG | - | - | - | - | 0.247 | 0.275 | 0.430 | 0.429 | 0.373 | 0.398 | 0.383 | 0.396 | 0.278 | 0.322 |

Table 8: Ablation results of the latest model 'TimePFN' with mixed paradigm. Since the model only has open-sourced pretrained checkpoints with 96-step prediction horizon instead of full training scripts, all our ablation experiments are conducted in inference mode. The Hadamard product and addition maintained the main accuracy of attention without training.

| Models | TimePFN | | | | | |
|---|---|---|---|---|---|---|
| | SA | | Hadam Prod | | Addition | |
| Metric | MSE | MAE | MSE | MAE | MSE | MAE |
| Traffic | 1.108 | 0.613 | 1.088 | 0.597 | 1.043 | 0.620 |
| Electricity | 0.315 | 0.384 | 0.310 | 0.378 | 0.399 | 0.473 |
| Weather | 0.210 | 0.255 | 0.224 | 0.265 | 0.203 | 0.253 |
| ETTh1 | 0.453 | 0.440 | 0.493 | 0.457 | 0.472 | 0.452 |
| ETTh2 | 0.329 | 0.363 | 0.337 | 0.366 | 0.323 | 0.362 |
| ETTm1 | 0.638 | 0.513 | 0.663 | 0.512 | 0.688 | 0.533 |
| ETTm2 | 0.212 | 0.291 | 0.223 | 0.300 | 0.216 | 0.297 |
| Solar | 0.941 | 0.731 | 1.404 | 0.880 | 0.884 | 0.700 |

Table 9: Comparison results between MB-MLP and 3 essential paradigms. It maintains the main accuracy of self-attention.

| Models | | MB-MLP (Inf) | | Informer | | MB-MLP (Pat) | | PatchTST | | MB-MLP (iTr) | | iTransformer | |
|---|---|---|---|---|---|---|---|---|---|---|---|---|---|
| Metric | | MSE | MAE | MSE | MAE | MSE | MAE | MSE | MAE | MSE | MAE | MSE | MAE |
| Traffic | 96 | 0.595 | 0.348 | 0.614 | 0.350 | 0.387 | 0.266 | 0.361 | 0.247 | 0.403 | 0.289 | 0.356 | 0.258 |
| | 192 | 0.604 | 0.362 | 0.618 | 0.352 | 0.401 | 0.271 | 0.380 | 0.255 | 0.428 | 0.304 | 0.375 | 0.268 |
| | 336 | 0.624 | 0.368 | 0.637 | 0.358 | 0.413 | 0.277 | 0.393 | 0.263 | 0.441 | 0.310 | 0.388 | 0.274 |
| | 720 | 0.636 | 0.378 | 0.636 | 0.367 | 0.443 | 0.295 | 0.432 | 0.286 | 0.468 | 0.324 | 0.421 | 0.289 |
| | AVG | 0.615 | 0.364 | 0.626 | 0.357 | 0.411 | 0.277 | 0.392 | 0.263 | 0.435 | 0.307 | 0.385 | 0.272 |
| Electricity | 96 | 0.214 | 0.322 | 0.214 | 0.319 | 0.135 | 0.230 | 0.130 | 0.222 | 0.142 | 0.240 | 0.133 | 0.228 |
| | 192 | 0.220 | 0.325 | 0.273 | 0.369 | 0.150 | 0.244 | 0.148 | 0.240 | 0.160 | 0.256 | 0.153 | 0.248 |
| | 336 | 0.232 | 0.338 | 0.228 | 0.332 | 0.166 | 0.261 | 0.165 | 0.259 | 0.176 | 0.273 | 0.173 | 0.268 |
| | 720 | 0.262 | 0.362 | 0.220 | 0.323 | 0.205 | 0.294 | 0.210 | 0.298 | 0.215 | 0.304 | 0.208 | 0.302 |
| | AVG | 0.232 | 0.337 | 0.234 | 0.336 | 0.164 | 0.257 | 0.163 | 0.255 | 0.173 | 0.268 | 0.167 | 0.262 |
| Weather | 96 | 0.182 | 0.239 | 0.180 | 0.238 | 0.163 | 0.212 | 0.156 | 0.205 | 0.153 | 0.203 | 0.163 | 0.211 |
| | 192 | 0.231 | 0.277 | 0.217 | 0.268 | 0.207 | 0.252 | 0.195 | 0.242 | 0.198 | 0.245 | 0.207 | 0.251 |
| | 336 | 0.271 | 0.307 | 0.270 | 0.305 | 0.256 | 0.290 | 0.250 | 0.284 | 0.249 | 0.285 | 0.256 | 0.291 |
| | 720 | 0.347 | 0.359 | 0.351 | 0.362 | 0.324 | 0.338 | 0.320 | 0.335 | 0.323 | 0.337 | 0.326 | 0.337 |
| | AVG | 0.258 | 0.296 | 0.255 | 0.293 | 0.237 | 0.273 | 0.230 | 0.267 | 0.231 | 0.268 | 0.238 | 0.273 |
| ETTh1 | 96 | 0.468 | 0.465 | 0.395 | 0.421 | 0.406 | 0.427 | 0.382 | 0.405 | 0.520 | 0.488 | 0.405 | 0.419 |
| | 192 | 0.447 | 0.462 | 0.423 | 0.437 | 0.442 | 0.445 | 0.414 | 0.421 | 0.561 | 0.509 | 0.454 | 0.450 |
| | 336 | 0.469 | 0.471 | 0.444 | 0.451 | 0.459 | 0.453 | 0.431 | 0.435 | 0.548 | 0.511 | 0.472 | 0.466 |
| | 720 | 0.524 | 0.510 | 0.483 | 0.487 | 0.515 | 0.503 | 0.449 | 0.466 | 0.556 | 0.529 | 0.549 | 0.530 |
| | AVG | 0.477 | 0.477 | 0.436 | 0.449 | 0.456 | 0.457 | 0.419 | 0.432 | 0.546 | 0.509 | 0.470 | 0.466 |
| ETTh2 | 96 | 0.369 | 0.407 | 0.382 | 0.412 | 0.293 | 0.351 | 0.274 | 0.336 | 0.341 | 0.387 | 0.307 | 0.363 |
| | 192 | 0.411 | 0.439 | 0.405 | 0.435 | 0.352 | 0.388 | 0.339 | 0.379 | 0.392 | 0.419 | 0.390 | 0.412 |
| | 336 | 0.390 | 0.431 | 0.380 | 0.425 | 0.336 | 0.387 | 0.331 | 0.381 | 0.374 | 0.416 | 0.419 | 0.433 |
| | 720 | 0.422 | 0.453 | 0.419 | 0.446 | 0.382 | 0.424 | 0.379 | 0.421 | 0.416 | 0.444 | 0.417 | 0.444 |
| | AVG | 0.398 | 0.432 | 0.397 | 0.430 | 0.341 | 0.387 | 0.331 | 0.379 | 0.381 | 0.416 | 0.383 | 0.413 |
| ETTm1 | 96 | 0.342 | 0.388 | 0.399 | 0.425 | 0.312 | 0.353 | 0.292 | 0.343 | 0.303 | 0.354 | 0.306 | 0.360 |
| | 192 | 0.374 | 0.405 | 0.422 | 0.439 | 0.348 | 0.378 | 0.331 | 0.369 | 0.342 | 0.378 | 0.345 | 0.382 |
| | 336 | 0.419 | 0.432 | 0.448 | 0.455 | 0.377 | 0.397 | 0.365 | 0.392 | 0.374 | 0.398 | 0.378 | 0.402 |
| | 720 | 0.495 | 0.470 | 0.526 | 0.509 | 0.424 | 0.428 | 0.419 | 0.425 | 0.435 | 0.431 | 0.444 | 0.440 |
| | AVG | 0.408 | 0.424 | 0.449 | 0.457 | 0.365 | 0.389 | 0.352 | 0.382 | 0.363 | 0.390 | 0.368 | 0.396 |
| ETTm2 | 96 | 0.219 | 0.298 | 0.239 | 0.307 | 0.175 | 0.266 | 0.165 | 0.255 | 0.185 | 0.265 | 0.172 | 0.265 |
| | 192 | 0.287 | 0.341 | 0.319 | 0.354 | 0.230 | 0.302 | 0.220 | 0.292 | 0.253 | 0.311 | 0.249 | 0.314 |
| | 336 | 0.313 | 0.360 | 0.341 | 0.380 | 0.286 | 0.338 | 0.278 | 0.329 | 0.301 | 0.344 | 0.294 | 0.345 |
| | 720 | 0.428 | 0.426 | 0.485 | 0.453 | 0.369 | 0.390 | 0.367 | 0.385 | 0.376 | 0.391 | 0.375 | 0.394 |
| | AVG | 0.312 | 0.356 | 0.346 | 0.374 | 0.265 | 0.324 | 0.258 | 0.315 | 0.279 | 0.328 | 0.273 | 0.330 |

