# OpenReview forum: "Is the Attention Matrix Really the Key to Self‑Attention in Multivariate Long‑Term Time Series Forecasting?"
_ICLR.cc/2026/Conference — ICLR 2026 Conference Withdrawn Submission_

### Official Review · Reviewer_WHwv · 2025-10-25

**Soundness:** 3
**Presentation:** 3
**Contribution:** 2
**Rating:** 4
**Confidence:** 5

**Summary:**

This paper systematically investigates the importance of the attention mechanism in Transformer-based forecasting models. Its findings challenge a fundamental premise of the architecture by demonstrating that the presence or absence of the attention mechanism has a negligible impact on model performance.

**Strengths:**

The authors conducted a series of intriguing experiments demonstrating that the attention mechanism contributes little to time series forecasting performance, which challenges prevailing assumptions in the field. The conclusion is further strengthened by the extensive evaluations across multiple model architectures, lending considerable support to their claim.

**Weaknesses:**

1. In Figure 5, why does the single-branch configuration (branch number = 1) achieve the best overall performance? This observation appears to contradict the view that multi-branch structures are essential for capturing diverse temporal patterns.

2. The experimental analysis remains somewhat superficial. For instance, how would performance be affected if the attention matrix is replaced with a random matrix, while removing all Hadamard products and matrix additions? Furthermore, what is the impact of more sophisticated embedding strategies on the results?

3. While the authors highlight the limited role of attention, they do not provide in-depth analysis regarding the underlying reasons for its ineffectiveness, nor do they suggest potential alternative mechanisms or solutions.

**Questions:**

Given that the attention mechanism is shown to be non-essential, what is the role of positional encoding in this context? Was positional encoding incorporated in the models, and if so, how does it affect the performance when attention is absent or replaced?

---

### Official Review · Reviewer_RMkb · 2025-10-28

**Soundness:** 2
**Presentation:** 2
**Contribution:** 1
**Rating:** 2
**Confidence:** 4

**Summary:**

This paper challenges the conventional belief that the dynamic attention matrix is the key to self-attention's success in multivariate long-term time series forecasting (MLTSF). Through extensive experiments on six different Transformer models, the authors demonstrate that this matrix-based token interaction can be replaced by simple element-wise operations, such as the Hadamard product or addition, while maintaining or even improving performance.

The paper's central hypothesis is that the effectiveness of self-attention in this task originates not from the attention matrix, but from the multi-branch feature extraction architecture provided by the parallel Query, Key, and Value (QKV) projections. To validate this claim, the authors construct and propose a MB-MLP model that isolates this specific structure.

**Strengths:**

1. It challenges existing research assumptions and provides a novel perspective on the role of self-attention in MLTSF.

2. It demonstrates that computationally efficient and simple modifications (element-wise operations), which remove the core token-interaction mechanism of attention, result in negligible performance degradation.

**Weaknesses:**

1. While the paper questions self-attention, its own results show that the original self-attention mechanism still performs better in several cases. This is particularly noticeable for strong baseline models like PatchTST and iTransformer, especially on high-dimensional datasets (e.g., Traffic, Electricity). Since these models are SOTA, this performance gap weakens the paper's central claim.

2. The paper's primary conclusion is that the proposed simpler methods are "almost as good as" self-attention, rather than demonstrably superior. This ambiguity about the practical benefit (why replace it if it's not better?) can be perceived as a lack of strong motivation or practical novelty.

3. The core idea that parts of self-attention are unnecessary is not entirely new. It invites direct comparison with the NeurIPS paper "Are self-attentions effective for time series forecasting?" (which this paper also cites), which previously demonstrated that removing self-attention from PatchTST led to less performance loss and even gains in some cases. This paper's findings could be seen as an incremental extension of this existing work, raising concerns about its novelty.

4. The presentation of results, such as in Figure 3, makes direct comparisons difficult. The performance margins are often small, and the varying scales of MSE/MAE across different datasets make it hard to intuitively grasp the practical significance of the differences. Furthermore, the paper's structure, which jumps directly into experiments (Chapter 3) to disprove attention before formally proposing the MB-MLP, can feel disjointed and may obscure the novelty of the MB-MLP model itself.

**Questions:**

Please refer to weaknesses

---

### Official Review · Reviewer_mJqK · 2025-10-30

**Soundness:** 2
**Presentation:** 2
**Contribution:** 1
**Rating:** 4
**Confidence:** 3

**Summary:**

This paper analyzes the effectiveness of the attention matrix in multivariate long-term time series forecasting (MLTSF). The authors explore three central research questions:

1. Is the attention matrix truly the key to self-attention in MLTSF?
2. Does the performance of self-attention primarily stem from multi-branch feature mapping and fusion?
3. Is the attention matrix worth the computational cost?

Each of these questions is highly relevant to the field of time series forecasting, which encompasses various approaches such as DLinear, Patch-TST, and CATS. The authors attempt to address these questions through small-scale experiments and empirical analyses, replacing components in the model to examine their relative impact.

**Strengths:**

The paper introduces a validation model, the Multi-Branch MLP (MB-MLP), which adopts an MLP-based multi-branch architecture similar to DLinear. This approach is interesting and may provide a new perspective for understanding feature fusion in time series models. The empirical comparisons and ablation studies could potentially inspire further research in this area.

**Weaknesses:**

(Major Issues)

The effectiveness of self-attention in time series forecasting has already been investigated in recent studies such as Zeng et al. (2023) and Kim et al. (2024). These works were well-received due to their strong empirical evidence showing that competitive time series models can be developed without self-attention mechanisms. Their structural novelty was sufficient to validate the hypothesis that the existence of a good time series forecasting model without self-attention.

In contrast, this paper primarily relies on ablation experiments to support its claims. As a result, the conclusions are difficult to interpret. For instance, in Figure 3, the MSE and MAE differences between self-attention (SA) and other variants are not substantial, despite the authors’ claim that “the Hadamard version leads to a 2.59% decrease in MSE and a 1.71% decrease in MAE.” Such differences may not be statistically or practically significant enough to support the conclusion that self-attention is inefficient. Similar issues appear across other ablation results.

The proposed MB-MLP model is conceptually interesting, yet the paper lacks comprehensive experimental evidence to substantiate its benefits. The degraded performance of MB-MLP suggests that multi-branch structures alone can be insufficient for achieving SOTA performance in time series forecasting. More importantly, the paper would benefit from deeper empirical and theoretical analyses. If the authors want to claim that “Figure 4(c) and (d) prove that MB-MLP can reproduce the main body,” they should support this assertion with theoretical reasoning or statistical validation, such as theoretical bounds or significance testing.

Overall, the manuscript reads more like a technical report or workshop paper rather than a thorough, peer-reviewed study. To strengthen the contribution, the authors should consider expanding their experiments to include additional datasets and benchmark models.

(Minor Issues)

- Several figures (e.g., Figures 1 and 4) are difficult to interpret due to inconsistent performance scales across datasets. The authors should normalize or reorganize the visualizations to improve readability.
- The manuscript appears to have been written in haste and contains several typographical and stylistic errors. For example:
“The success of attention-free models like DLinear Zeng et al. (2023)” should be written as “The success of attention-free models like DLinear (Zeng et al., 2023).”

**Questions:**

Refer to Weaknesses.

---

### Official Review · Reviewer_M3Ni · 2025-10-30

**Soundness:** 3
**Presentation:** 2
**Contribution:** 3
**Rating:** 6
**Confidence:** 3

**Summary:**

The paper argues that in multivariate long-term time-series forecasting (MLTSF) the attention matrix is not the main source of Transformer performance. The authors replace dot-product attention with element-wise ops across six Transformer baselines and report largely comparable accuracy and build a minimalist multi-branch MLP (MB-MLP) that keeps parallel Q/K/V projections and fuses them with simple ops, also matching baselines closely.

**Strengths:**

- The analysis is model-agnostic, including Time-token (Informer-like), patch-token (PatchTST), variate-token (iTransformer), plus recent hybrids (ICTSP, Leddam, TimePFN).
- MB-MLP retaining only QKV branches matches most of PatchTST/iTransformer performance with far less compute, which supports the authors' claim.
- Table 2 suggests attention matrix adds non-trivial time/memory over Hadamard while offering marginal average gains.

**Weaknesses:**

- The paper introduces a “Hadamard operation–based attention” as a structural ablation. However, similar element-wise multiplicative or formulations have been previously explored in related works, such as some linear attention mechanisms, gated attention, etc. The authors are encouraged to discuss them.

- While many curves overlap, there are some exceptions, suggesting the attention matrix can act as a stabilizer in some regimes. Is it possible to analyze when and why failures happen?

- The paper cites DLinear / PatchMLP / FITS / CATS, but should more explicitly contrast what they proved vs what this paper proves (e.g., attention-free vs. attention-matrix-free but QKV-kept). Right now, novelty is present but under-argued.

- Some baselines are constrained (e.g., ICTSP scripts only for ETT; Leddam limited to horizon 96 on large datasets), which may bias aggregate conclusions. Please summarize where coverage is partial and provide sensitivity analyses.

**Questions:**

- When do ablations fail? Provide dataset/horizon characteristics predicting when element-wise fusion underperforms (e.g., high inter-token dependency scenarios?).
- Why do authors choose the Hadamard operation?
- Do you think Softmax/nonlinearity plays a very important role in token mixing?

---

### Note · Authors · 2025-12-08

I have read and agree with the venue's withdrawal policy on behalf of myself and my co-authors.